# Long-Term Immuno-Response and Risk of Breakthrough Infection After SARS-CoV-2 Vaccination in Kidney Transplantation

**DOI:** 10.3390/vaccines13060566

**Published:** 2025-05-26

**Authors:** Vincenzo Bellizzi, Mario Fordellone, Carmine Secondulfo, Paolo Chiodini, Giancarlo Bilancio

**Affiliations:** 1Division of Nephrology, Sant’Anna e San Sebastiano Hospital, 81100 Caserta, Italy; 2Medical Statistics Unit, University of Campania Luigi Vanvitelli, 80138 Naples, Italy; mario.fordellone@unicampania.it (M.F.); paolo.chiodini@unicampania.it (P.C.); 3Department of Medicine, Surgery and Dentistry, University of Salerno, 84081 Baronissi, Italy; csecondulfo@unisa.it (C.S.); gbilancio@unisa.it (G.B.); 4Division of Nephrology, San Giovanni di Dio e Ruggi d’Aragona University Hospital, 84081 Salerno, Italy

**Keywords:** kidney transplant, COVID, SARS-CoV-2, mRNA vaccine, BNT162b2, hybrid immunity, immune response, infection risk

## Abstract

**Background**: Kidney transplant (KTx) recipients exhibit impaired responses to SARS-CoV-2 vaccination. Correlates of vaccine-induced immunity and risk factors for breakthrough infection are not fully defined. This study evaluated the humoral response trajectories and determinants of breakthrough infection in KTx recipients. **Methods**: KTx recipients received two doses of the BNT162b2 mRNA vaccine three weeks apart and a booster after six months. Patients were categorized based on pre-vaccination status: previous COVID-19 disease (DIS), asymptomatic SARS-CoV-2 infection (INF), or infection-naïve (NEG). Serum anti-spike antibody titers were assessed at baseline, before the second dose, and at 1, 3, 6, 9, and 12 months. Linear mixed models and survival analyses were performed. **Results**: Of 326 enrolled patients, 189 with complete time-point data were included in the longitudinal analysis. Antibodies were detectable in 89% of DIS/INF at baseline and 91% before the second dose, but were negligible in NEG. In NEG, the seropositivity increased after vaccination and booster, reaching 78% at 12 months. Age (−5% per year, *p* < 0.001) and BMI (+10% per unit, *p* = 0.004) influenced titers; antimetabolites and steroids had strong negative effects (−70%, *p* = 0.005; −84%, *p* = 0.001). Breakthrough infections occurred in 104 (31.9%); 40% were asymptomatic, and 2 patients died. An mTOR inhibitor was associated with a reduced infection risk (OR 0.27 [CI: 0.09–0.70], *p* = 0.009). Higher antibody titers correlated with delayed infection (*p* = 0.063). **Conclusions**: In KTx patients, humoral response to SARS-CoV-2 vaccination is limited in infection-naïve patients but improved by booster dosing; the hybrid immunity is more effective. Immunosuppressive regimens influence the immune response, and mTOR inhibitors may protect against breakthrough infection.

## 1. Introduction

In immunodeficient patients, the response to vaccines is often inadequate, or at least not as strong as that in the general population, and kidney transplant recipients are no exception to this rule, mainly because of immunosuppressive therapy; the most well-known example of this phenomenon is the reduced response to the vaccine against hepatitis B [1]. On the other hand, in solid organ transplants, the high prevalence of comorbidities, such as diabetes mellitus, hypertension, cardiovascular diseases, chronic lung disease, and obesity, plays a negative role in susceptibility to virus infections [2,3].

Vaccines against SARS-CoV-2 revealed a reduced humoral response rate in immunocompromised patients, such as those with hematological or solid cancers [4], and these patients were excluded from the SARS-CoV-2 vaccine trials [5]. Kidney transplant (KTx) recipients, while at high risk of SARS-CoV-2 infection, have an impaired humoral immune response to SARS-CoV-2 vaccination [6]. Several reports, most with small sample sizes and limited follow-up, show that the protective antibody production is reduced and the seroconversion rate is low [6,7]; hence, the vaccine efficacy seems impaired, and these patients are prone to more frequent SARS-CoV-2 infections [8] and more severe disease [9,10]. Other reports in solid organ transplant recipients show a decline of protective antibody concentration six months after vaccination [11], thus making further interventions mandatory. To contrast such limitations, a booster vaccine dose has been administered in many countries to renal transplant subjects, and a larger humoral response, compared to a standard two-dose regimen, has been achieved [12], eliciting the appearance of neutralizing antibodies even in previously non-responsive patients [13].

Despite vaccination, KTx patients continue to face an increased risk of COVID-19. Moreover, these frail patients have a high risk of severe infection and the worst outcomes, including a disproportionately high incidence of hospitalization, acute respiratory failure, and mortality. This heightened vulnerability is largely attributable to the chronic immunosuppressive therapy required to maintain the graft function, which compromises both innate and adaptive immune responses. In addition, understanding the long-term immune response to mRNA vaccines (intensity, determinants, spread, and duration) and the implications of prior SARS-CoV-2 infection or disease before vaccination remains unclear. Moreover, different virus variants have emerged after the initial waves, and the incidence of SARS-CoV-2 infection has risen again worldwide; nonetheless, few data concerning the incidence, clinical severity, and protective factors of COVID breakthrough infection after the booster in KTx recipients are still available [14].

This study explored the long-term immune response and its correlates following complete mRNA vaccination for SARS-CoV-2. Additionally, it investigated the incidence, related factors, and clinical characteristics of COVID-19 breakthrough infections occurring after two vaccine doses and a booster shot.

## 2. Methods

This study was approved by the local Institutional Ethical Committee (n.78 RPSo, 8 April 2021, and DD n.52, 12 April 2021 from “Comitato Etico Campania Sud” ASL Napoli 3-Sud; cometicocampaniasud@aslnapoli3sud.it), and the enrolled patients signed an informed consent form.

A cohort of adult (age > 18 years) KTx patients with a transplant age of at least six months, stable renal function defined as no acute rejection in the previous six months, and no COVID-19 infection in the previous three months were enrolled. All patients received two doses of the BNT162b2 mRNA vaccine, three weeks apart, in an in-hospital facility in March 2021 and were invited to receive a booster dose after six months.

Before and during the vaccination cycle, subjects underwent clinical evaluations and blood and urine measurements within the scheduled visits for the usual clinical follow-up [15]. Subsequently, they were closely surveyed to discover new COVID-19 through a PCR SARS-CoV-2 RNA assay or antigenic test, undertaken soon after any symptoms or contact with infected or suspected individuals.

The SARS-CoV-2 anti-spike IgG titer (SARS-CoV-2 IgG II, ABBOTT; Architect 5 i1000SR, ABBOTT Diagnostics, Abbott Park, IL, USA: sensitivity = 98.3–95.0%, CI: 90.6–100%, specificity = 99.5–95.0%, CI: 97.1–100%; cut-off for positivity threshold (correspondent to the protection threshold) of antibody (Ab) level ≥ 50 AU/mL; conversion factor from AU to BAU = 0.142 × AU/mL) [16] was measured before the first vaccine dose (V1), at the time of administration of the second dose (V2), and during the follow-up at months 1 (M1), 3 (M3), 6 (M6), 9 (M9), and 12 (M12) after the 2nd dose (time-points covariate). Serum samples were sealed and stored in a dedicated, 24 h monitored refrigerator at −20 °C.

To identify subjects with hidden SARS-CoV-2 infections before vaccination, we conducted a serum assay test for anti-nucleocapsid antibodies against SARS-CoV-2 in all participants (Elecsys Anti-SARS-CoV-2 test—ROCHE; Cobas e411 ROCHE Diagnostics, Totkreuz, Switzerland; sensitivity = 99.5–95%, CI: 97.0–100%; specificity = 99.5–95%, CI: 98.6–99.8%) [17]. Patients who demonstrated high titers of either anti-nucleocapsid or anti-spike antibodies were classified as previously infected. From this assessment, we categorized kidney transplant (KTx) patients into three groups based on their SARS-CoV-2 status (the patient-status covariate): (1) **Disease (DIS)**: Subjects with a clinical history and symptoms or a molecular diagnosis (swab test) of COVID-19. (2) **Infection (INF)**: Subjects without a known clinical history of COVID-19 but who tested positive for anti-nucleocapsid antibodies or had high anti-spike antibody titers before vaccination (nobody received prior vaccination, and serum samples were always collected before vaccination). (3) **Negative (NEG)**: Subjects with no clinical history of COVID-19 and who did not show any anti-nucleocapsid antibodies (anti-N Ab) or anti-spike antibodies (anti-S Ab). The local Institutional Ethics Committee approved the study, and all patients provided informed consent by signing the necessary forms.

### Statistics and Analyses

Continuous variables were reported as the mean and standard deviation or the median and interquartile ranges (IQRs) according to their distribution. Categorical variables were reported as absolute frequencies and percentages. Differences in the baseline characteristics of the DIS, INF, and NEG groups were tested by ANOVA or Kruskal–Wallis tests for continuous variables, and Pearson’s chi-squared or Fisher’s exact tests for categorical variables. To estimate the subjects’ protected proportion by *patient-status*, the interval estimation by Clopper–Pearson [18] was applied.

A linear mixed model (LMM) was performed to estimate the covariates’ longitudinal effect on the antibody titer. The mixed effects regression model uses all available data and can properly account for the correlation between repeated measures. The covariates included in the model were *patient status* and *time-points* as categorical variables and their interaction. Missing data, such as absent antibody titer measurements at certain time-points, were handled using the default approach of LMM, which performs a complete-case analysis (listwise deletion). In particular, the model was assessed using only the available data without any imputation, under the assumption that data are missing at random (MAR). As a result, each time-point contributes to the model estimation based on the observed data at that time-point, without requiring all participants to have data at every time-point. To address the issue of multiple comparisons, we performed post hoc pairwise comparisons using Tukey’s Honestly Significant Difference (HSD) test [19] applied to the fitted LMM. This approach includes adjustment for multiple testing and provides family-wise error rate control.

For this analysis, only the NEG patient group was considered, and then only the *time-point* covariate was used for the statistical model estimate.

The same LMM analysis was also performed to estimate the effect of baseline clinical characteristics during the administration of the second dose of vaccine at V2 and their interaction on the log 10-transformed antibody titer; for this analysis, only the NEG group was considered, and then only the time-point covariate was used to adjust clinical variable estimates.

To analyze the effects of clinical characteristics on the SARS-CoV-2 infection risk, logistic regression models according to individual characteristics (adjusted for age, gender, BMI, previous transplant, transplant type, age at transplant, diabetes, antimetabolite, mTORi, steroid, eGFR), were evaluated at 3 time-points: completion of the vaccine cycle (V2) and 3 and 6 months after vaccination (M3, M6). Finally, at the same three time-points, the Kaplan–Meier approach was used to measure the time to infection for the groups with antibody titer levels under and over the protective cut-off of 50 AU/mL. The differences between groups were tested by the log-rank test.

To evaluate the impact of the humoral status on the following infections, the last available serum anti-S Ab level prior to the onset of COVID-19 in KTx recipients who had an infection was compared with the anti-S Ab level of other patients at the time-point closest to the time-point identified for the infected patient. Statistical tests with *p*-values smaller than 0.05 were considered statistically significant.

All statistical analyses were performed with the R Studio Statistical software, version 4.1.3.

## 3. Results

### 3.1. Patients

A total of 326 kidney transplant (KTx) patients met the inclusion criteria and were followed up during the study, while 189 patients—22 DIS, 10 INF, and 157 NEG—underwent antibody level measurements at all time-points and entered the longitudinal trajectory analyses (Table 1).

### 3.2. Immune Response

At baseline (V1), the anti-S Ab titer was above the positivity threshold in 89% [CI: 71–98] of the DIS/INF group (*n* = 27). At the time-point V2, 91% [CI: 75–98] of these latter subjects (*n* = 32) were above the positivity threshold, while among the NEG group (*n* = 157), the prevalence of Ab-positive patients was 9% [CI: 5–15]. At the time-point M1 (1 month after the second vaccine dose), the prevalence of anti-S Ab above the positivity threshold was 93% [CI: 78–99] in DIS/INF (*n* = 30); in NEG (*n* = 213), it reached 50% [CI: 43–57] of subjects. At the time-point M3 (3 months after 2nd vaccine dose), the anti-S Ab was quite stable in all groups (*n* = 32 for DIS/INF and 204 for NEG). From M6 (six months after the second dose), which is the time of booster administration, the anti-S Ab stayed stable in the DIS/INF group (*n* = 28); conversely, in NEG (*n* = 265), the anti-S Ab started an increasing trend, with 57% [CI: 51–63], 74% [CI: 51–63], and 78% [CI: 71–84] of patients with an anti-S Ab titer over the positivity threshold at 6, 9, and 12 months after the second dose, respectively (Figure 1).

Statistically significant differences in antibody titer among *time-points* (*p* < 0.0001) within *patient status*, and *patient status* within *time-points* (*p* < 0.0001), were detected. Among the NEG group, there were differences in the anti-S Ab titer among all the time-points (*p* < 0.003) except between V1 and V2 (*p* = 0.32), M1 and M3 (*p* = 0.89), M1 and M6 (*p* = 0.14), and M9 and M12 (*p* = 0.12); within DIS, there were differences in the anti-S Ab titer among V1 versus all the others time-points (*p* < 0.0001) only; within INF, there were no differences in the anti-S Ab titer. Within the *patient status* at all time-points, at V1–M6, there were differences in Ab serum levels between the NEG and DIS/INF groups (*p* < 0.0001); whereas, at time-points M9 and M12, there were differences only between NEG and DIS (*p* < 0.0003) (Figure 1).

In the 157 patients in the NEG group, the impact of the booster and reinfection on the anti-S Ab titer was explored in a separate analysis. At M6 (six months after the second vaccine dose), a few patients refused the booster (NEG, *n* = 8; INF, *n* = 2; DIS, *n* = 1), and their anti-S Ab serum titer declined, with the prevalence of anti-S Ab-positive subjects falling below 50%. Among the NEG patients who received the booster, the antibody titer was raised, with the prevalence of subjects with the anti-S Ab titer above the positivity threshold reaching 75%. The response was even higher in subjects who had breakthrough infections after the booster, for both anti-S Ab levels and the prevalence of anti-S Ab-positive subjects, reaching 90% (Figure 2).

### 3.3. Determinants of Immune Response

Table 2 displays the results regarding the relationship between individual characteristics and antibody titer during vaccine administration in NEG patients. All of the time-point coefficients were positive and statistically significant (*p* < 0.0001) with the biggest increasing effect on the anti-S Ab titer at M1 (1460% to V2) and M12 (2817% to V2). A marginal but significant effect on the anti-S Ab titer was due to age, with a decreasing impact of 4.9% (*p* < 0.0001) for each year of age increase, and due to BMI, which showed an increasing impact of 10.4% (*p* = 0.004) for each unit of BMI increase; the immunosuppressive drugs, i.e., antimetabolites and steroids showed a marked negative effect on the anti-S Ab titer of 70.4% (*p* = 0.005) and 84% (*p* = 0.001), respectively (mTOR and calcineurin inhibitors were not included in the model for a very high statistically significant inverse association with antimetabolites and steroids).

### 3.4. Breakthrough Infection

During the follow-up, 104 KTx recipients developed SARS-CoV-2 infection after M6; 13% of the infections occurred during the Delta variant period (September–November 2021), and 87% ensued during the Omicron wave (February–April 2022). Active screening for breakthrough SARS-CoV-2 infections following vaccination was not systematically performed in all patients. To identify the onset of infection, patients were advised to carefully monitor any symptom possibly related to COVID-19 as well as any personal contact with known infected people, to refer it to the transplant center, and a swab was soon performed in all these conditions to assess the infection.

Patients with breakthrough SARS-CoV-2 infection after vaccination had similar characteristics to patients without infection, except mTORi therapy which was significantly higher in non-infected patients. KTx patients that received the booster were close to 90% in both groups; among the infected, 87 (83.6%) received the booster before infection, 4 (3.8%) received it after infection, 3 (2.9%) received the booster but skipped the anti-S Ab measurement, and 10 (9.6%) never received a booster. Two infected KTx recipients (1.9%) required hospitalization and died (Table 3). Among KTx recipients with breakthrough infection, 40% were asymptomatic and all patients (symptomatic or not) had similar characteristics.

In the risk analyses of SARS-CoV-2 breakthrough infection, the positive antibody titer, individual characteristics (age, gender, and BMI), and disease status (eGFR and diabetes) were not associated with the risk of infection; among immunosuppressors, no relation was found for steroids and antimetabolite drugs, while mTORi significantly reduced the risk by 73% (OR 0.266 [CI: 0.094–0.695], *p* = 0.009) at M6. The patient status did not affect this analysis (Figure 3). In patients with breakthrough infection, the presence of anti-S Ab above the positivity threshold at M3 and M6 was not associated with breakthrough infection incidence; however, a protective anti-S Ab level is correlated with a delay in the onset of infection; the median onset of infection, indeed, was 1 month later for anti-S Ab-positive subjects (Figure 4).

## 4. Discussion

This study evaluates the long-term immune and clinical responses to the BNT162b2 mRNA vaccination against SARS-CoV-2 in a large cohort of kidney transplant patients. In infection-naïve patients, only 50% and 75% reach positive antibody levels after the two doses of the vaccination and the booster, respectively. In contrast, concerning the protection of patients who received the vaccination but had undergone a previous infection, regardless of the presence of clinical symptoms, their so-called “hybrid immunity” was complete. Individual and disease characteristics have a marginal effect or no effect on determining the antibody response. Antimetabolites and steroids have a major negative impact, while the use of mTORi and CNI may facilitate a humoral response. Notably, the risk of breakthrough infection after vaccination with two doses plus the booster is not associated with individual and clinical characteristics or antibody titer, while mTORi and any level of neutralizing antibodies above the positive threshold seem protective, being associated with the delay of the onset of infection.

In kidney-transplanted patients naïve to SARS-CoV-2 infection, the humoral response to the mRNA vaccine resulted in impaired intensity, protective efficacy, and persistence. This is not new. Previous pieces of evidence have already described incomplete and low-titer responses to the third [20,21] and fourth [22] vaccine doses, leaving many patients unprotected. A more recent, large report in a similar cohort under a similar full vaccination program already showed that a large proportion of kidney transplant patients remained seronegative against SARS-CoV-2 [23]. Furthermore, a large systematic review evaluating the immune response to COVID-19 vaccines in solid organ transplants (SOTs) concluded that a booster enhances the immunogenicity of vaccines, but many SOT recipients remain unprotected [24]. Overall, in this setting of vulnerable patients, the role of hybrid immunity [25] remains to be understood. A natural infection should induce a wider immunological reaction as compared to a vaccine’s unique surface antigen, and hence provide better protection. Combined natural and vaccine protection (hybrid immunity) could be more effective. This study provides novel additional information on the comparison between vaccine immunity (patients naïve to SARS-CoV-2) and hybrid immunity (previous infection or disease patients plus vaccine) and on their long-term behavior. The antibody trajectories had a similar trend, though they were largely under-placed in infection-naïve patients in which the antibodies at their zenith reached a lower level than that at the nadir in hybrid immunity; the prevalence of positive levels of neutralizing antibodies, ranging between 49 and 78% along the follow-up, as compared against a rage 88 to 100%, respectively. Notably, the anti-S Ab level decreases strikingly below the positivity threshold after 6 months when the booster is skipped.

These data are original and add further knowledge regarding the vaccine response and the role of SARS-CoV-2 hybrid immunity in renal transplant recipients. The humoral response to the mRNA vaccine in infection-naïve kidney transplant patients is impaired, but not in those KTx recipients who have contracted COVID-19; this seems to exclude insufficient stimulation by mRNA vaccines [26], suggesting that this result is due to different antigen stimulation between the natural virus and the vaccine [22,27], and also highlights the role of hybrid immunity in transplant patients.

The key role of immunosuppression in the impaired humoral response to COVID-19 vaccination in kidney-transplanted patients has already been underlined [6,28,29]. Several original reports [30,31,32,33,34] and two large systematic reviews and meta-analyses [35,36] pointed out the strongest, independent association of mycophenolate (MMF) and corticosteroids with a negative humoral response in solid organ transplantation. One small report in kidney transplant patients alone showed that MMF was associated with the lowest humoral response [37]. To our knowledge, this is the largest study exploring the relationship between different immunosuppressant drugs and antibody response after vaccination in kidney transplants alone. Here, it has been shown that antimetabolites and steroids are the major determinants of impaired humoral response; meanwhile, CNI and mTORi moved in opposite directions, likely playing a protective role in the antibody response. The effect of mTORi against a low antibody response has been previously hypothesized in a meta-analysis of 29 studies in SOT [35]. In a further study in a cohort of SOT, however, it was suggested that such an association is not independent, but rather confounded by MMF [38]. Here, we suggest through a robust linear mixed model predictive analysis that mTORi and CNI are protective against a weak humoral response to vaccines in kidney transplant patients. In addition, this study confirms a mild protective role of body mass index and a negative impact of older age in the setting of KTx recipients, as already shown in comprehensive SOT [30,32,34,35,36].

In this study, the incidence of COVID-19 breakthrough infections in kidney transplant patients after SARS-CoV-2 vaccination was 31.9% in the whole cohort. The breakthrough infection arose 4.1 [IQR 3.6–4.0] months after the booster. Most infections (27% out of the overall 32%) occurred during the wave of Omicron variants, BA.1 (most common), BA.1.1, and BA.2, which were characterized by high infectivity [39]; indeed, Omicron can extensively escape BNT162b2-elicited neutralization [40]. In a large cohort of over half a million people in the US, including persons with and without immune deficiency, the breakthrough infection rate was 5 per 1000 person-months after full vaccination, being almost twice that in SOT individuals [41]. Moreover, in a prospective primary care cohort of 14 million people who had received the full vaccination across all UK, the severe COVID-19 outcome rate was 0.2% (corresponding to 7.6 events per 1000 person-years), the individuals either under immunosuppression or with chronic kidney disease or kidney transplant being at a higher adjusted rate ratio risk (aRR: 5.8 [5.53–6.09], 3.71 [2.9–4.74] and >10.0, respectively) [42]. In our cohort, around 60% of infected KTx recipients had symptoms of a mild degree; two patients, 0.61% of all patients, entered the full vaccination program, and 1.92% of those infected had severe COVID-19 outcomes, with hospitalization and death. Overall, given the differences between populations, the incidence of either breakthrough infection or severe COVID-19 is similar to that observed elsewhere.

Infected KTx patients had no significant differences from those without infection, except for the lower prevalence of mTORi, which agrees with the stronger immune protection associated with this immunosuppressive agent in KTx recipients [43]. Indeed, mTORi was associated with higher protection against the onset of infection in KTx patients (Figure 3). In contrast, the level of neutralizing antibodies and the prevalence of protected subjects were not different; hence, in this study, the anti-S Ab titer did not play a protective role against the onset of breakthrough infection in the predictive analysis. Previous studies in other immunosuppressed patients showed opposite results on the protective role of anti-S Ab [14,32]. In addition, among solid organ transplant patients (including kidney), non-high anti-S Ab levels were associated with a higher risk of infection [44]. Also, in a cohort of patients affected by lymphoma, there was a direct correlation between the anti-S Ab level after vaccination and infection, with a 13-fold lower infection risk for anti-S Ab over the positivity threshold [45]. In a large cohort of SOTs, the breakthrough infection rate was 25.2%, and older age, mycophenolate, and corticosteroids were independent risk factors for severe breakthrough infection; also in this study, however, the antibody response and the severe infections themselves were very heterogenous among transplant types, and the authors did not conclude on the role of anti-S Ab protection in the risk of infection in KTx recipients [46]. Finally, a recent paper discovered a close relationship between low anti-spike IgG levels after vaccination and severe breakthrough infection in kidney transplant recipients [47]; in this study, the prevalence of protected patients was similar to our study, but the overall infection rate was lower (22.3%), while the severe infection was much higher (18.8% of those infected). Hence, the role of the anti-S Ab titer in reinfection remains uncertain.

Of interest, in this study, the antibody level above the positivity threshold was associated with a delay in the onset of SARS-CoV-2 infection in kidney transplant patients (Figure 4). This finding is original and remarkable, possibly suggesting a correlation between the presence of detectable antibody titers and increased resistance to infection in this specific setting of frail patients. The one-month delay in the onset of infection is noteworthy, as it could have significant implications for the clinical management of patients. The marginal significance at M3 (*p* = 0.063), though not strong, may indicate a trend worthy of further investigation and could have clinical implications for the management of transplant patients concerning SARS-CoV-2 infection. Delaying SARS-CoV-2 infection in kidney transplant recipients may offer clinical advantages. A longer time to infection allows patients to benefit from heightened vigilance during subsequent waves of virus circulation, potentially increasing adherence to protective behaviors such as self-isolation, mask-wearing, and early symptom reporting. Moreover, if hospitalization becomes necessary, delayed infection may coincide with a period following the peak of an infection wave, when healthcare systems are less strained and medical resources, including intensive care unit capacity and antiviral treatments, are more readily available. This temporal shift could contribute to improved clinical outcomes in this high-risk population.

Overall, available data are conflicting, with none reporting a protective effect of mTORi and the impact of antibodies on breakthrough infection, except for the severe form, being uncertain. The present study provides new information on the predictive role of mTORi against SARS-CoV-2 breakthrough infection and on the protective effectiveness of anti-S Ab levels to delay the onset of infection. This information is even more relevant since breakthrough infections after prior immunization or infections are becoming increasingly common, especially after the rise in novel variants [48].

On the whole, the clinical behavior of breakthrough COVID-19 infections has been favorable, and this may be due to the lower clinical morbidity of the current virus variants, but also to the clinical effectiveness of the immunization regimen as a result of the BNT162b2 mRNA vaccine against severe, critical, or fatal COVID-19. This effect could be related to any antibody level above the positive threshold, which is somehow efficacious, as testified by the delay in the onset of infection; this is in agreement with the general population, where higher levels are more protective [49].

A strength of this work is that it provides an altogether comprehensive, long-term picture of the immune response to the vaccine, determinants of the immune response, comparisons between infection-naïve or previously infected patients, so-called hybrid immunity, and the impact of breakthrough infection after vaccination including its determinants in the same quite large cohort of a unique solid organ transplant (kidney) and over quite a long period of time. Such information may help us to understand how to better improve more efficacious strategies for the future.

A limitation of this study is that no vaccine other than the BNT162b2 mRNA vaccine has been administered to the patients, and thus it is unknown whether a mixed-scheme vaccination could better improve the humoral response in renal transplant patients. Also, this study is single-center; hence, geographical, ethnic, or social factors could have affected COVID-19 diffusion. Furthermore, the subgroups DIS and INF are of a relatively small size, possibly limiting the power of the statistical analysis. Moreover, all enrolled patients had a transplant age of over 1 year, and information on vaccine efficacy in de novo kidney transplants is lacking. Another possible limitation is that the incidence of infection after vaccination was not systematically screened, but was investigated only after contact with infected subjects or symptom onset; some individuals considered not to be infected could have unknowingly contracted a SARS-CoV-2 infection. Due to the observational nature of this study, it is not possible to infer the causality relation between mTORi therapy and its associated effects on vaccine response and infection protection.

## 5. Conclusions

In conclusion, this study comprehensively evaluated both vaccine immunity and hybrid immunity after full vaccination against COVID-19 in a cohort of kidney-transplanted patients, showing that the humoral response in infection-naïve patients (vaccine immunity) is impaired and is lower compared to the higher and long-lasting protection due to hybrid immunity. Moreover, antimetabolites and steroids are the major determinants of impaired humoral responses. This information could be useful for future approaches to vaccination in immunocompromised subjects against other possible future viral diseases.

## Figures and Tables

**Figure 1 vaccines-13-00566-f001:**
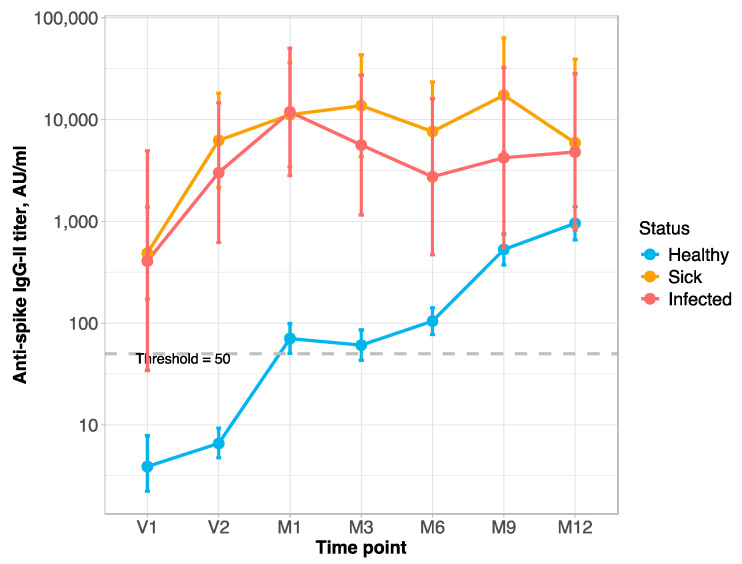
The behavior of titer and prevalence of protective level of anti-spike Ab across *patient status* and *time-points* during the study. (**Top**): Line-plot titer distribution (points represent the mean values of titer in the specific group/time-point, error-bars represent 95%-CI; dashed line is the threshold for protective anti-S Ab level). (**Middle**): Box-plots Ab titer distribution (bold lines represent the median value; dashed line is the threshold for protective Ab level). (**Bottom**): Prevalence of positive anti-S Ab titer (95%-CI by Clopper–Pearson). **Time-points**: Administration of vaccine: V1 = before 1st dose; V2 = before 2nd dose. M1, M3, M6, M9, and M12 = 1, 3, 6, 9, and 12 months after 2nd dose; M9 and M12 = 3 and 6 months after booster. **Box-plot**: Anti-S Ab titers are on the log_10_ scale.

**Figure 2 vaccines-13-00566-f002:**
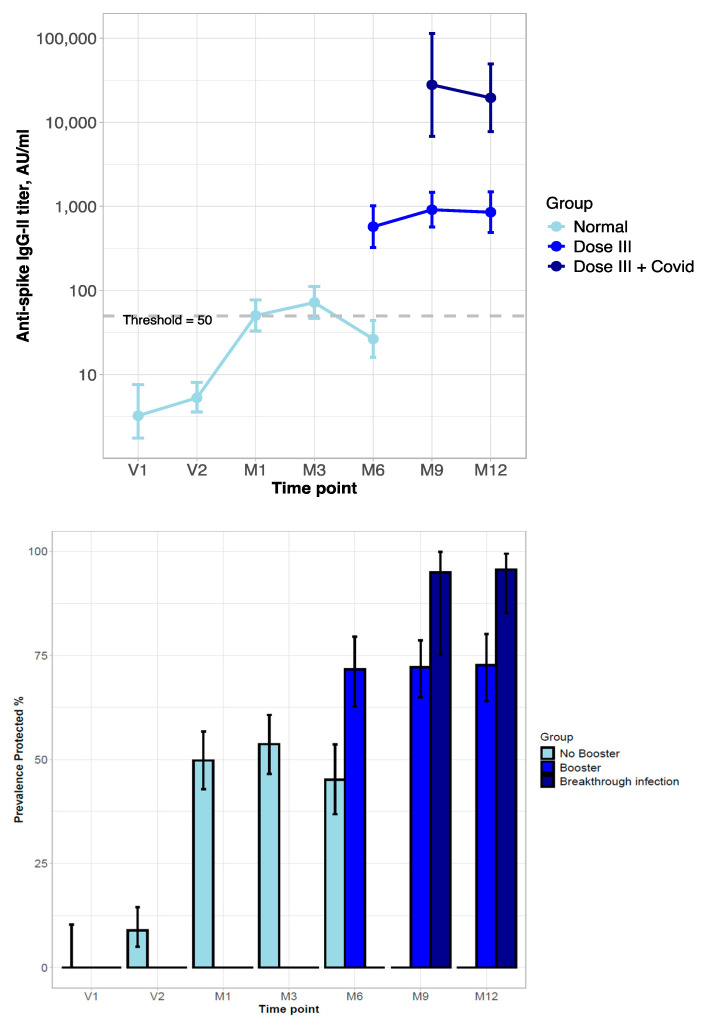
Behavior of titer and prevalence of protective level of anti-spike Ab in kidney transplant patients never infected by SARS-CoV-2 (NEG group) during vaccination and after booster and/or COVID-19 infection. (**Top**): Line-plot titer distribution (error bars represent 95%-CI; dashed line is the threshold for protective anti-S Ab level). (**Bottom**): Prevalence of positive anti-S Ab titer of neutralizing SARS-CoV-2 Ab (with 95%-CI estimated by Clopper–Pearson). Time-points: Administration of vaccination cycle (V1, V2), 1, 3, and 6 months post-vaccine (booster administration) (M1, M3, M6), and 9 and 12 months post-vaccine (M9, M12).

**Figure 3 vaccines-13-00566-f003:**
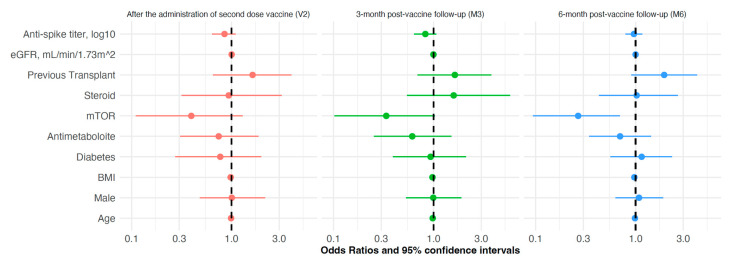
Forest plot analysis of the risk of SARS-CoV-2 infection in kidney transplant patients after anti-SARS-CoV-2 mRNA vaccination, according to individual characteristics (adjusted for factors listed in Table 3) at different time-points along the follow-up: completion of vaccine cycle (V2, red color), 3- and 6-month post-vaccine follow-up (M3, M6, green and blue colors, respectively).

**Figure 4 vaccines-13-00566-f004:**
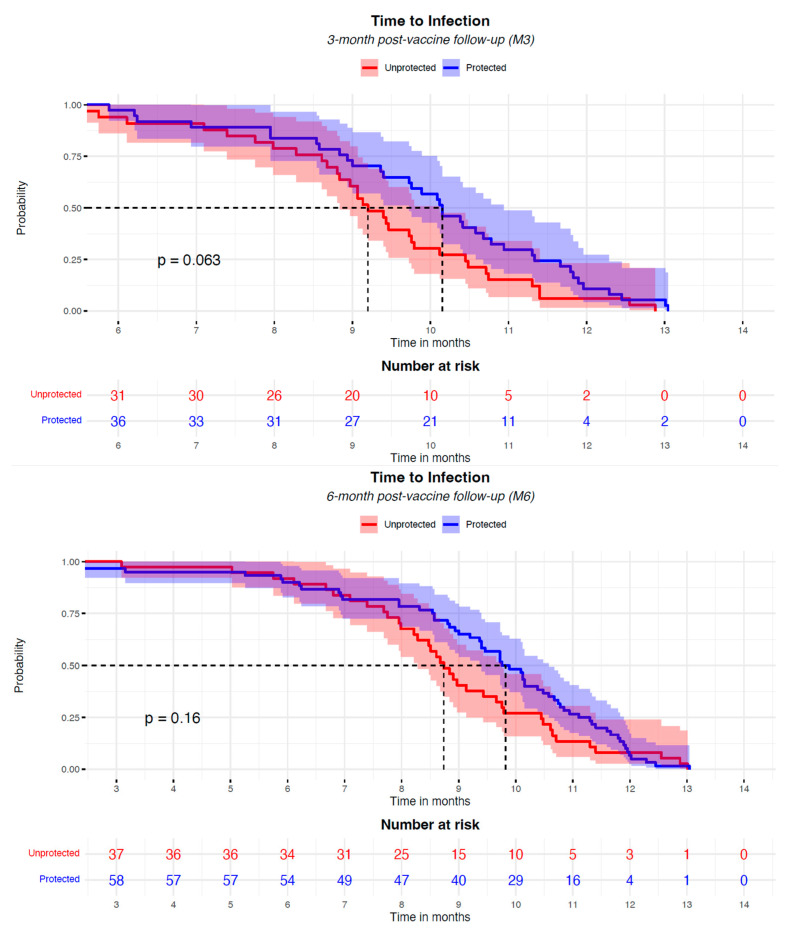
The Kaplan–Meier analysis of the risk of COVID-19 infection in KTx patients after the anti-SARS-CoV-2 mRNA vaccination, according to the presence of a positive level of neutralizing Ab, started at the 3-month and 6-month post-vaccine time (M3, M6). Participants were classified as “protected” (blue curve) or “unprotected” (red curve) based on the presence of a positive neutralizing antibody titer at 3 months and 6 months after the two vaccine doses. The Kaplan–Meier curves have been aligned, starting the plotting from 6 months and 3 months after the full vaccination for the two analyses, respectively. No infection events were recorded before these time-points. The booster administration occurred in the timeframe between 3 and 6 months after the vaccination. Dashed lines represent the median time to infection of the groups.

**Table 1 vaccines-13-00566-t001:** Baseline characteristics of groups according to SARS-CoV-2 status, for patients who underwent the repeated measures analysis for immune response to the vaccine.

	All Patients	SARS-CoV-2 Subgroups		
		Disease	Infection	Negative
Patients, *n*	189	22	10	157
Gender, % males	73.5	77.3	50.0	74.5
Age, years	55.4 ± 11.1	52.0 ± 14.0	49.9 ± 13.1	56.2 ± 12.8
BMI, kg/m^2^	26.2 ± 4.6	27.0 ± 5.0	26.1 ± 4.4	26.1 ± 4.7
Diabetes, %	16.4	4.5	20.0	17.8
Kidney disease, %				
Glomerulonephritis	15.3	27.3	20.0	13.4
Diabetic nephropathy	2.6	0.0	0.0	3.2
Interstitial nephropathy	1.1	0.0	0.0	1.3
Nephroangiosclerosis	4.8	9.1	0.0	4.5
Immune diseases	11.6	4.5	10.0	12.7
Unknown	33.9	36.4	50.0	32.5
Congenital/ADPKD	30.7	22.7	20.0	32.5
Previous transplant, %	15.9	9.1	0.0	17.8
Current transplant, %				
Deceased donor	87.8	90.8	90.0	87.3
Living donor	10.6	9.1	10.0	10.8
Kidney/pancreas	1.6	0.0	0.0	1.9
Transplant age, years	12.9 ± 8.22	13.0 ± 8.37	17.0 ± 8.38	12.6 ± 8.17
eGFR^EPI^, mL/min/1.73 m^2^	61.7 ± 27.9	65.0 ± 30.9	56.0 ± 31.7	61.6 ± 27.4
Immunosuppressive drugs, %				
Calcineurin-*i*	97.4	95.5	100	97.5
mTOR-*i*	16.4	13.6	10.0	17.2
Antimetabolite	69.3	81.0	60.0	68.2
Steroids	89.9	95.5	100	97.5

**Table 2 vaccines-13-00566-t002:** Changes in antibody titers from baseline (V2) to 12 months (M12) at the linear mixed model analysis.

Time-Points from V2 to M12				
Coefficient	Value	%	95% CI	*p*-Value
(Intercept)	1.956	8936.50	417.6/157,298.3	0.002
M1	1.193	1459.60	420.0/4588.1	<0.001
M3	0.839	590.20	123.9/2023.2	0.001
M6	0.870	641.30	153.5/2067.7	<0.001
M9	1.317	1974.90	557.7/6461.5	<0.001
M12	1.465	2817.40	769.0/9672.4	<0.001
Age	−0.022	−4.9	−7.3/−2.5	<0.001
Gender, male	−0.021	−4.7	−53.5/95.4	0.896
BMI	0.043	10.40	3.0/18.0	0.004
Antimetabolite	−0.528	−70.4	−87.2/−31.0	0.005
Steroid	−0.792	−83.9	−94.2/−55.1	0.001
Donor type, living donor	0.162	45.20	−42.3/264.8	0.426
Diabetes	−0.222	−40.0	−74.4/40.6	0.237
eGFR	0.000	0.00	−1.4/1.6	0.945

The reference time-point is V2. Coefficients are reported on the log_10_ scale. The “Coefficient value%” column represents the estimated percentage change in antibody levels relative to V2, based on back-transformed coefficients (i.e., fold-change from log_10_ to original scale). V2 = 2nd dose administration, M1, M3, M6, M9, and M12 = 1, 3, 6, 9, and 12 months after 2nd dose.

**Table 3 vaccines-13-00566-t003:** Characteristics of patients with or without breakthrough infection after vaccination.

	COVID-19 Breakthrough Infection	No COVID-19 Breakthrough Infection	*p*-Value
Patients, *n* (%)	104 (31.9)	222 (68.1)	
Age, years	55.1 ± 12.9	56.9 ± 11.7	0.22
Gender, % male	66.3	69.8	0.52
BMI, kg/m^2^	26.0 ± 4.2	26.8 ± 4.7	0.20
Previous transplant, %	16.3	11.7	0.29
Transplant type, % alive	14.4	10.8	0.36
Age at transplant, years	42.2 ± 13.2	44.2 ± 13.5	0.15
Diabetes, %	15.4	17.1	0.75
Antimetabolite, %	75.0	69.4%	0.36
mTORi, %	8.7	17.6	0.04
Steroid, %	91.3	89.2	0.69
eGFR, mL/min/1.73 m^2^	62.0 ± 22.0	62.0 ± 28.7	0.93
Anti-spike IgG-II titer V2, AU/mL	5.55 ± 23.30	4.10 ± 40.70	0.88
Anti-spike IgG-II titer V2, % >50 AU/mL	10.6	14.4	0.57
Anti-spike IgG-II titer M3, AU/mL	65.30 ± 461.00	115.00 ± 951.00	0.23
Anti-spike IgG-II titer M3, % >50 AU/mL	35.6	45.0	0.25
Booster, %	90.4	96.4	0.11
Anti-spike IgG-II titer M6, AU/mL	193.00 ± 1160.00	140.00 ± 3690.00	0.81
Anti-spike IgG-II titer M6, % >50 AU/mL	57.7	53.2	0.80
Symptomatic, *n* (%)	63 (60.5)	---	0.99
Death, *n* (%)	2 (2)	---	0.99

## Data Availability

Data described in the manuscript will be made available upon request by application to the Medical Statistics Unit of the University of Campania Luigi Vanvitelli of Naples, Italy.

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
