# Peer review of "Long-Term Immuno-Response and Risk of Breakthrough Infection After SARS-CoV-2 Vaccination in Kidney Transplantation"

_vaccines, 2025, doi:10.3390/vaccines13060566_

Round 1
Reviewer 1 Report
Comments and Suggestions for Authors
This clinical study is interesting and well presented.
However some remarks have to be made.
line 85-87: one “(“ missing; the so-called positive threshold correspond to protection one.
Line 103:“show any anti-nucleocapsid or anti-SARS-CoV-2“; N (nucleocapsid) and S (spike) are both SARS-Cov-2 proteins. Text to be modified.
Line 179: “a few patients refused the booster” ; how many ? The number of patients in the 3 groups should be indicated in figure 2 and 3.
Line 246: “the protection of patients have had previous infection or disease” ; they were both infected, symptomatic or not!
Line 270-271: “Ab level striking decreases “ ; strikingly ? ; “after 6 months whether (if?, when?) the booster...”
Line 281: MMF not explicited
Line 316: “resulted in highly protective against...” a name missing
For clarity, all over the text, the two immune responses against these two antigens should be differentiated and do not limit it to “ab” ; e.g. anti-S-ab and anti-N-ab.
There is not a difference between DIS and INF patients as both group consist of patients which have been infected by SARS-Cov-2 perhaps after an aymptomatic form. The ones with isolated anti-S antibodies may have been previously vaccinated. To be confirmed.
How the diagnosis of breakthrough infection was obtained in patients? Was it performed systematically ? Which technique(s) was (were) used?
Figure 4 is barely understandable. Some additional indications would be welcomed. It represent the follow-up of the 104 fully vaccinated patients which presented a breakthrough infection according to their serological status which should be indicated in the legend. Do the survey began after the 2 vaccine doses? When was the booster administered.
N.B.: even if it is not fully protective, a natural infection should induce a wider immunological reaction as compared to a vaccinal unique surface antigen, hence a better protection.
Author Response
Q1) This clinical study is interesting and well presented. However, some remarks have to be made.
A1) We thank the reviewer for the positive comments. A point-by-point reply follows.
Q2) 1) line 85-87: one “(“ missing; the so-called positive threshold corresponds to protection one.
A2) We thank the reviewer for pointing out the missing; it has been filled.
Q3) Line 103: “show any anti-nucleocapsid or anti-SARS-CoV-2“; N (nucleocapsid) and S (spike) are both SARS-CoV-2 proteins. Text to be modified.
A3) We thank the reviewer for pointing out this imprecision; it has been corrected in the manuscript
Q4) Line 179: “a few patients refused the booster”; how many? The number of patients in the 3 groups should be indicated in Figures 2 and 3.
A4) Thank you for this suggestion. In the revised version of the manuscript, we added the number of patients who refused the booster and the number of patients in the 3 groups in the “Immune response” section
Q5) Line 246: “the protection of patients who have had previous infection or disease”; they were both infected, symptomatic or not!
A5) We thank the reviewer for highlighting this confused, unclear statement, a more appropriate terminology has now been provided
Q6) Line 270-271: “Ab level striking decreases”; strikingly? ; “after 6 months, whether (if?, when?) the booster...”
A6) We thank the reviewer for pointing out the typo and giving suggestions to improve clarity. The text has been corrected.
Q7) Line 281: MMF not explicated
A7) We thank the reviewer for pointing out the missing explanation for this abbreviation; this point has been addressed.
Q8) Line 316: “resulted in highly protective against...” a name is missing
A8) We thank the reviewer for highlighting this missing; the text has been modified as suggested.
Q9) For clarity, all over the text, the two immune responses against these two antigens should be differentiated and do not limit it to “ab” ; e.g. anti-S-ab and anti-N-ab.
A9) We thank the reviewer for the suggestions; this point has been addressed throughout the paper.
Q10) There is not a difference between DIS and INF patients as both group consist of patients which have been infected by SARS-Cov-2 perhaps after an aymptomatic form. The ones with isolated anti-S antibodies may have been previously vaccinated. To be confirmed.
A10) We thank the reviewer for their thoughtful comment. In our study, the DIS and INF groups were distinguished based on the presence or absence of symptoms, a clinically relevant variable in this particularly vulnerable patient population. Individuals with isolated anti-S antibodies had never received prior vaccination, as serum samples were collected immediately before the start of the vaccination campaign. In Italy, during the COVID-19 pandemic, vaccine distribution was strictly regulated, and vaccines were only administered through official vaccination hubs under close supervision. This point has been further clarified in the text, Lines 109-110
Q11) How the diagnosis of breakthrough infection was obtained in patients? Was it performed systematically? Which technique(s) was (were) used?
A11) We thank the reviewer for addressing this point. Section “3.4. Breakthrough Infection” has been updated with this information. COVID-19 screening was not performed systematically; the appearance of any possible symptom related to SARS-CoV-32 infection was carefully monitored, and when this was the case, the patients underwent the test.
Q12) Figure 4 is barely understandable. Some additional indications would be welcomed. It represents the follow-up of the 104 fully vaccinated patients which presented a breakthrough infection according to their serological status which should be indicated in the legend. Do the survey began after the 2 vaccine doses? When was the booster administered.
A12) Many thanks for this observation point that allows us to better clarify this relevant point. The Kaplan Mayer analysis of the risk of COVID-19 infection in KTx patients after the anti-SARS-CoV-2 m-RNA vaccination, according to the presence of a positive level of neutralizing Ab, started at the 3-month and 6-month post-vaccine time (M3, M6). Specifically, participants were classified as "protected" (blue curve) or "unprotected" (red curve) based on the presence of a positive neutralizing antibody titer at 3 months and 6 months after the two vaccine doses; the Kaplan-Meier curves have been aligned, starting the plotting from 6 months and 3 months after the full vaccination for the two analyses, respectively. No infection events were recorded before these time points; the booster administration occurred in the timeframe between 3 and 6 months after the vaccination
Q13) N.B.: even if it is not fully protective, a natural infection should induce a wider immunological reaction as compared to a vaccinal unique surface antigen, hence a better protection.
A13) We agree with the reviewer, and this hypothesis is mentioned in the discussion section; it has now been implemented according to the suggestion: Lines 475-477.
Reviewer 2 Report
Comments and Suggestions for Authors
1. Abstract (Lines 10–33)
Consider restructuring the abstract to more clearly delineate background, methods, results, and conclusions for better readability and alignment with journal standards.
2. Introduction (Lines 37–69)
The introduction effectively outlines the problem but could benefit from a more detailed rationale for selecting kidney transplant patients and specifying the knowledge gap this study addresses.
3. Methods (Lines 70–139)
Provide more detail on how missing data were handled during analysis (e.g., missing antibody titers at time points).
Clarify if adjustments were made for multiple comparisons in the mixed model post-hoc tests.
4. Cohort Description (Lines 141–148)
Table 1 is comprehensive; however, p-values indicating non-significance should be interpreted cautiously, especially with small subgroup sizes (e.g., INF, n=10).
5. Immune Response (Lines 149–185)
Include confidence intervals in the text for NEG group post-booster response at M9 and M12 to be consistent with earlier time points.
Figure 1 is informative; consider increasing font size and clarity of legends for publication.
6. Determinants of Immune Response (Lines 195–209)
Table 2 should clearly specify the time-point reference (V2) in the caption and explain whether “Coefficient value %” refers to change on the log10-transformed scale or back-transformed.
7. Breakthrough Infections (Lines 210–231)
Highlight the timeframe during which infections occurred relative to variant prevalence more explicitly in the results section.
Consider adding a multivariate model including time-since-booster to evaluate risk more thoroughly.
8. Discussion (Lines 242–364)
The discussion is generally strong. However, the distinction between association and causality should be more clearly stated when referring to mTORi effects.
Consider briefly discussing the clinical implications of delayed infection in Ab-positive patients.
9. Limitations (Lines 365–373)
Good acknowledgment of limitations. Consider mentioning the relatively small numbers in certain subgroups (DIS, INF) as this may limit statistical power for interaction effects.
10. Formatting and Consistency
Ensure consistent use of abbreviations (e.g., “Ab,” “mTORi”) throughout.
Some line spacing and punctuation are inconsistent, particularly around percentages and units (e.g., “4.3% (p=0.004)” vs “4.3% (p = 0.004)”)—standardize this for clarity.
11. References (Lines 425–533)
Overall, references are appropriate and current. Ensure all web-based citations are functional and archived, particularly reference 17.
Author Response
- Abstract (Lines 10–33)
Q1. Consider restructuring the abstract to more clearly delineate background, methods, results, and conclusions for better readability and alignment with journal standards.
A1. We thank the reviewer for the suggestion; the abstract was modified as suggested
- Introduction (Lines 37–69)
Q2. The introduction effectively outlines the problem but could benefit from a more detailed rationale for selecting kidney transplant patients and specifying the knowledge gap this study addresses.
A2. We thank the reviewer for the suggestion; the introduction has been modified to enhance clarity
- Methods (Lines 70–139)
Q3. Provide more detail on how missing data were handled during analysis (e.g., missing antibody titers at time points). Clarify if adjustments were made for multiple comparisons in the mixed model post-hoc tests.
A3. Thank you for your comments. “Missing data, such as absent antibody titer measurements at certain time points, were handled using the default approach of LMM, which performs a complete-case analysis (listwise deletion). In particular, the model was performed using only the available data without any imputation, under the assumption that data are missing at random (MAR). As a result, each time point contributes to the model estimation based on the observed data at that time point, without requiring all participants to have data at every time point. To address the issue of multiple comparisons, we performed post-hoc pairwise comparisons using Tukey's Honest Significant Difference (HSD) test19 applied to the fitted LMM. This approach includes adjustment for multiple testing and provides family-wise error rate control.” We have added this information to the Statistical Analysis section of the revised manuscript for clarity.
- Cohort Description (Lines 141–148)
Q4. Table 1 is comprehensive; however, p-values indicating non-significance should be interpreted cautiously, especially with small subgroup sizes (e.g., INF, n=10).
A4. We fully agree with this comment, and in this revised version of the paper, we deleted the p-values column from Table 1 to avoid any misinterpretations.
- Immune Response (Lines 149–185)
Q5. Include confidence intervals in the text for NEG group post-booster response at M9 and M12 to be consistent with earlier time points
A5. Thanks for asking. The %[CI] for the post-booster anti-S Ab titer over the positivity threshold in the NEG group is 74% [CI:51-63] at M9 and 78% [CI:71-84] at M12, and is reported in the “immune response” section
Q6. Figure 1 is informative; consider increasing font size and clarity of legends for publication.
A6. Thank you very much for this suggestion. In the revised version of the manuscript, the quality of figures has been improved, and the font size has been enlarged
- Determinants of Immune Response (Lines 195–209)
Q7. Table 2 should clearly specify the time-point reference (V2) in the caption and explain whether “Coefficient value %” refers to change on the log10-transformed scale or back-transformed.
A7. Thank you for highlighting this important point. LMM was referred to log₁₀-transformed antibody titers. We have to apologize; indeed, we realized that the previous version of the paper, the percentage change values reported in the Table were mistakenly calculated using the natural logarithm (loge) rather than log₁₀. We have now corrected these values using the appropriate back-transformation for log₁₀; both the direction and significance of the results do not change vs the previous version of the paper. We have also revised the caption of Table 2 to explicitly state that V2 (baseline) is the reference time point used in the model. Furthermore, we clarified that the “Coefficient value%” refers to back-transformed estimates, representing the relative change in antibody titers compared to V2 on the original (non-log-transformed) scale. The “Coefficient” column reports the estimate on the log₁₀ scale from the LMM, while the “%” column shows the corresponding percentage change, calculated as (10coefficient−1)×100, to facilitate interpretation. The manuscript has been updated accordingly.
- Breakthrough Infections (Lines 210–231)
Q8. Highlight the timeframe during which infections occurred relative to variant prevalence more explicitly in the results section. Consider adding a multivariate model including time-since-booster to evaluate risk more thoroughly.
A8. We thank the reviewer for this observation. We did not perform a multivariate analysis based on the time-since-booster, since most of the patients, either had or not the infection, received the booster and, mainly, such vaccine dose was administered within a short time period (around one month) in all patients; hence there would be no difference according to the booster administration
- Discussion (Lines 242–364)
Q9. The discussion is generally strong. However, the distinction between association and causality should be more clearly stated when referring to mTORi effects.
A9. We agree with the reviewer; as this study has an observational design, causality cannot be implied in the aforementioned observations. This limitation has now been added to the discussion and the limitation section.
Q10. Consider briefly discussing the clinical implications of delayed infection in Ab-positive patients.
A10. We thank the reviewer for this suggestion; a concise discussion of this aspect was added to the text
- Limitations (Lines 365–373)
Q11. Good acknowledgment of limitations. Consider mentioning the relatively small numbers in certain subgroups (DIS, INF) as this may limit statistical power for interaction effects.
A11. We thank the reviewer for the suggestion. This limitation has been added.
- Formatting and Consistency
Q12. Ensure consistent use of abbreviations (e.g., “Ab,” “mTORi”) throughout.
A12. Done.
Q13. Some line spacing and punctuation are inconsistent, particularly around percentages and units (e.g., “4.3% (p=0.004)” vs “4.3% (p = 0.004)”)—standardize this for clarity.
A13. Thank you for this suggestion. We standardised the line spacing and punctuation along the paper at our best.
- References (Lines 425–533)
Q14. Overall, references are appropriate and current. Ensure all web-based citations are functional and archived, particularly reference 17.
A14. We confirm that web-based citations are functional. The link in ref #17 leads to the webpage of the Elecsys® Anti-SARS-CoV-2, as intended.